# Altitudinal Variation Influences Soil Fungal Community Composition and Diversity in Alpine–Gorge Region on the Eastern Qinghai–Tibetan Plateau

**DOI:** 10.3390/jof8080807

**Published:** 2022-07-30

**Authors:** Jian Chen, Zuomin Shi, Shun Liu, Miaomiao Zhang, Xiangwen Cao, Miao Chen, Gexi Xu, Hongshuang Xing, Feifan Li, Qiuhong Feng

**Affiliations:** 1Key Laboratory of Forest Ecology and Environment of National Forestry and Grassland Administration, Ecology and Nature Conservation Institute, Chinese Academy of Forestry, Beijing 100091, China; cafchenjian@163.com (J.C.); liushun89@163.com (S.L.); z411412115@163.com (M.Z.); caoxiangwen67@163.com (X.C.); chenmiaocc@163.com (M.C.); xugexi@163.com (G.X.); xhs10062246@163.com (H.X.); 19139507805@163.com (F.L.); 2Miyaluo Research Station of Alpine Forest Ecosystem, Lixian County 623100, China; 3Co-Innovation Center for Sustainable Forestry in Southern China, Nanjing Forestry University, Nanjing 210037, China; 4Institute for Sustainable Plant Protection, National Research Council of Italy, 10135 Torino, Italy; 5Ecological Restoration and Conservation on Forest and Wetland Key Laboratory of Sichuan Province, Sichuan Academy of Forestry, Chengdu 610081, China; fqiuhong@163.com

**Keywords:** soil fungi, community composition, diversity pattern, trophic mode, soil pH, soil moisture, soil nutrients, altitude gradient

## Abstract

Soil fungi play an integral and essential role in maintaining soil ecosystem functions. The understanding of altitude variations and their drivers of soil fungal community composition and diversity remains relatively unclear. Mountains provide an open, natural platform for studying how the soil fungal community responds to climatic variability at a short altitude distance. Using the Illumina MiSeq high-throughput sequencing technique, we examined soil fungal community composition and diversity among seven vegetation types (dry valley shrub, valley-mountain ecotone broadleaved mixed forest, subalpine broadleaved mixed forest, subalpine coniferous-broadleaved mixed forest, subalpine coniferous forest, alpine shrub meadow, alpine meadow) along a 2582 m altitude gradient in the alpine–gorge region on the eastern Qinghai–Tibetan Plateau. Ascomycota (47.72%), Basidiomycota (36.58%), and Mortierellomycota (12.14%) were the top three soil fungal dominant phyla in all samples. Soil fungal community composition differed significantly among the seven vegetation types along altitude gradients. The α-diversity of soil total fungi and symbiotic fungi had a distinct hollow pattern, while saprophytic fungi and pathogenic fungi showed no obvious pattern along altitude gradients. The β-diversity of soil total fungi, symbiotic fungi, saprophytic fungi, and pathogenic fungi was derived mainly from species turnover processes and exhibited a significant altitude distance-decay pattern. Soil properties explained 31.27−34.91% of variation in soil fungal (total and trophic modes) community composition along altitude gradients, and the effects of soil nutrients on fungal community composition varied by trophic modes. Soil pH was the main factor affecting α-diversity of soil fungi along altitude gradients. The β-diversity and turnover components of soil total fungi and saprophytic fungi were affected by soil properties and geographic distance, while those of symbiotic fungi and pathogenic fungi were affected only by soil properties. This study deepens our knowledge regarding altitude variations and their drivers of soil fungal community composition and diversity, and confirms that the effects of soil properties on soil fungal community composition and diversity vary by trophic modes along altitude gradients in the alpine–gorge region.

## 1. Introduction

As decomposers, mutualists, or pathogens, soil fungi play an irreplaceable role in key ecological processes such as element biogeochemical cycles [1]. They drive the soil carbon (C) and nitrogen (N) cycle, mediate mineral nutrition for plants, and mitigate carbon limitations for other soil organisms [1,2]. Studying the effect of environmental changes on soil fungal community composition and diversity can enhance a comprehensive understanding of the entire ecosystem [3]. The α-diversity of soil fungi responds to climate change differently by trophic modes (i.e., symbiotroph, saprotroph, and pathotroph) [4,5], implying that the effect of different soil fungal trophic modes on ecosystem functioning needs to be explored in the future. Mountains provide an open, natural platform for studying how soil microbes respond to environmental change, especially climate change, at short altitude distances [6]. In contrast to soil bacteria [7,8], the effect of altitude on soil fungal community composition and diversity remains unclear [9], and it is also unclear whether altitude variations and their drivers of the soil fungal community vary by trophic modes [10].

Changes in vegetation types with altitude lead to significant changes in soil fungal community composition, as this is coupled tightly with vegetation properties through host specificity and the production of different organic substrates [11]. Soil fungal community composition varies significantly along altitude gradients [12,13,14]. Since the dominant taxa in soil fungi (taxonomic and trophic mode) exhibit different life strategies, determining the effect of altitude gradients on the dominant taxa may be important for interpreting the altitude patterns of fungal diversity and understanding ecosystem function [15]. Ascomycota and Basidiomycota are two common soil fungal dominant phyla, which show opposite distribution patterns along altitude [16]. The relative abundance of different soil fungal trophic modes shows different changes along altitude gradients: saprophytic fungi increase, while symbiotic fungi (mainly ectomycorrhizal fungi) do not change significantly [17].

Investigating the variation of soil fungal diversity with altitude could help to understand and predict the regulation of soil fungal diversity on ecosystem function in the context of climate change [18]. Diverse patterns have been found to exist in α-diversity of soil total fungi and trophic modes along altitude gradients [19,20,21], such as a monotonic increasing pattern [22,23], a monotonic decreasing pattern [24,25], no obvious pattern [26], a hump-shaped pattern [12,27], and a hollow distribution pattern [28]. The β-diversity is an important indicator for studying the relationship between soil microbial community composition and environmental factors [29,30]. More attention has been paid to the distribution patterns of soil bacterial β-diversity along altitude gradients than that of soil fungi [7,31]. Although a study found that soil fungal β-diversity was derived mainly from species turnover components [24], it remains unclear whether the main components of soil fungal β-diversity differ across different trophic modes along altitude gradients.

Edaphic properties may directly affect the soil fungal community composition along altitude gradients, more than climatic factors [32,33]. Changes in soil factors along altitude may produce environmental filtering that affects the soil fungal community composition [34], and its response to altitude is usually driven by soil moisture [35], soil temperature [36], soil pH [37], soil mineral nutrients [38], and the soil carbon to nitrogen ratio (C/N) [39], etc. The main drivers of soil fungal community composition along altitude gradients vary by trophic modes, and drivers for saprophytic fungi are affected mainly by soil pH, while pathogenic fungi are affected mainly by annual mean temperature [12]. Soil pH has been consistently reported as a critical factor affecting α-diversity of soil fungi along altitude gradients [28,40], but the relationship between soil pH and α-diversity of different soil fungal trophic modes is not identical, being correlated negatively with symbiotic fungi and correlated positively with saprophytic fungi and pathogenic fungi [12]. Some studies have also showed that soil total nitrogen and nitrate nitrogen are the key drivers for α-diversity of saprophytic and pathogenic fungi along altitude gradients, respectively [41]. The β-diversity of soil fungi was found to be driven by environmental factors, geographic distance, or their combination along altitude gradients [42,43,44]. The driver divergence of turnover and nestedness components of soil fungal β-diversity has been validated at regional scales [30], but it has not been validated along altitude gradients.

The Qinghai–Tibetan Plateau, known as the “Roof of the world”, is the highest plateau on earth. The alpine–gorge region on the eastern Qinghai–Tibetan Plateau has a large altitude span, and there is a complete altitudinal vegetation belt from the valley to the top of the mountain, which provides a natural platform for studying the altitude pattern of biodiversity [45]. Along the altitude gradient, the mid-altitude climatic conditions are more suitable for survival than the relatively harsh climatic conditions of dry valleys and alpine meadows at both ends of the gradient. The low-altitude (dry valleys) in the region is characterized by drought and higher temperature, caused by the foehn effect and topographical factors [46], whereas the high-altitude (alpine meadows) is characterized by lower temperature, stronger wind, and stronger ultraviolet radiation [47]. The altitude variations and their drivers of soil bacterial [8], total fungal [44], and ectomycorrhizal fungal [48] communities have been partially explored on the eastern Qinghai–Tibetan Plateau, but investigations to distinguish different fungal trophic modes are still lacking. To compare the altitude variations and their drivers of soil fungal communities of different trophic modes, we collected soil samples from valley to hilltop along a 2582 m altitude gradient in the alpine–gorge region to determine soil properties and soil fungal communities, based on fungal ITS sequences. We hypothesized that (i) the α-diversity of soil fungi (total and trophic modes) exhibited a distinct hump-shaped pattern along altitude gradients; (ii) the β-diversity of soil fungi (total and trophic modes) was derived mainly from species turnover processes along altitude gradients and the relative effects of environmental and geographic distance on β-diversity varied by trophic modes; (iii) the main drivers of soil fungal community composition varied by trophic modes.

## 2. Materials and Methods

### 2.1. Study Area

The study area is in the alpine–gorge region on the eastern Qinghai–Tibetan Plateau. Different vegetation types are distributed due to significant vertical climatic changes within the area (Figure 1). The mean annual temperature and mean annual precipitation are 5.3 ℃ and 750 mm, respectively (http://www.data.cma.cn (accessed on 1 March 2022)). A description of the sampling sites is shown in Table 1.

### 2.2. Plot Setup and Soil Sampling

In late July 2020, a total of 63 temporary plots were set up randomly and averagely in 7 different vegetation types along altitude gradients from 1663 m to 4245 m above sea level (Table 1). To avoid edge effects, all plots were set at the center of each vegetation type, at least 50 m from the upper and lower boundaries [15]. According to a literature review [9,28,48] and field investigation, 9 plots were set up in each vegetation type, and the plot area was determined to be 20 m × 20 m for forests, and 10 m × 10 m for shrubs and meadows. Plant community surveys were performed, and importance values were calculated to determine dominant plant species [49]. Geographic coordinates and altitude were recorded for each plot using a handheld GPS (Unistrong G130BD, Beijing, China). Within each plot, five evenly distributed sampling grids (1 m × 1 m) were set up and three soil cores (10 cm depth below the litter layer, 3.8 cm diameter) within each sampling grid were collected randomly. A total of 15 soil cores were combined as an individual sample, and then kept in sterilized plastic zipper bags. A total of 63 individual soil samples were collected, from low altitude to high altitude, within a week. Soil samples were transferred back to the laboratory in an icebox for subsequent assay analysis. The soil samples, after removing stones, animal residues, and plant residues and passing through a 2 mm sieve, were divided into two parts: one part was air-dried for the determination of soil physicochemical properties, and the other part was kept at −80 ℃ for DNA extraction.

### 2.3. Determination of Soil Physicochemical Properties

Soil water content (SWC) was determined as the amount of soil mass lost before and after drying (dried to constant weight at 105 °C) as a percentage of soil dry weight. Soil bulk density (BD) was determined as the ratio of the mass of oven-dried soil to the volume of the ring knife. Soil water-filled pore space (WFPS) was calculated according to the standard formula [50]. Soil conductivity (EC) was determined using a conductivity meter in a water–soil solution (*v/w*, 5:1). Soil pH was determined using a glass electrode meter (Mettler Toledo, Zurich, Swizerland) in a water–soil solution (*v/w*, 2.5:1). Soil organic carbon (SOC) was measured by the K_2_Cr_2_O_7_-external heating method. Soil total carbon (TC) and total nitrogen (TN) were measured using a Vario EL cube CHNOS Elemental Analyzer (Elementar Analysensysteme GmbH, Langenselbold, Germany). Soil total phosphorus (TP) and total potassium (TK) were determined by plasma emission spectrometer (Thermo IRIS Intrepid II XSP, Franklin, FL, USA), using the acid dissolution method (HNO_3_-HClO_4_-HF). Soil available phosphorus (AP) was determined by continuous flow autoanalyzer (SEAL AutoAnalyzer 3, Bran and Luebbe GmbH, Germany), using a molybdenum blue colorimetric method after extraction with a mixed solution (0.05 M HCl and 0.025 M H_2_SO_4_). Soil ammonium-N (NH_4_^+^-N) and nitrate-N (NO_3_^−^-N) were measured by continuous flow autoanalyzer (SEAL Auto Analyzer 3, Bran and Luebbe GmbH, Germany), after extraction with 2 M KCl solution. The soil carbon to nitrogen ratio (C/N), carbon to phosphorus ratio (C/P) and nitrogen to phosphorus ratio (N/P) were calculated based on TC, TN and TP.

### 2.4. DNA Extraction, MiSeq Sequencing, and Bioinformatics

A total of 63 individual soil samples of microbial DNA were extracted using a DNeasy PowerSoil Kit (100), following the manufacturer’s instructions. After the concentration of the extracted DNA was qualified, the fungal ITS2 genes were amplified using fITS7 (5’−GTGARTCATCGAATCTTTG−3’)/ITS4(5’−AGCCTCCGCTTATTGATATGCTTAART−3’). The PCR 25 μL reaction system includes 8.5 μL of deionized sterile water, 0.75 μL of each primer, 12.5 μL of KAPA enzyme, and 2.5 μL of DNA template. The PCR reactions were conducted using the following program: 3 min of denaturation at 95 °C, 35 cycles of 30 s at 98 °C, 30 s for annealing at 56 °C, and 30 s for elongation at 72 °C, and a final extension at 72 °C for 10 min. Deionized sterile distilled water was also used as a negative control in all PCR amplification steps to detect the presence of contamination during the experiment. PCR products were extracted from a 2% agarose gel and further purified using an OMEGA E.Z.N.A.® Cycle Pure Kit (OMEGA, USA) and quantified using QuantiFluor™-ST (Promega, Madison, WI, USA). Purified PCR products were subjected to high-throughput sequencing (IlluminaMiseq PE 250).

Raw sequences were demultiplexed, quality filtered by Trimmomatic, and merged by FLASH (version 1.2.11) [51]. The chimeric sequences were identified and removed using UCHIME [52], and the non-chimera sequences were screened for quality, and then operational taxonomic units (OTUs) were clustered with a 97% similarity cutoff using Usearch (version 11.0.667) [53]. The taxonomy of each ITS gene sequence was analyzed using the UNITE database [54], the confidence threshold was 0.65. The number of sequences per sample was normalized to the minimum sample size using the *sub.sample* command in Mothur (version 1.45.2) [55] to remove the effect of the number of sequences across samples on the fungal community. The raw reads have been deposited into the National Centre for Biotechnology Information (NCBI) Sequence Read Archive database (https://www.ncbi.nlm.nih.gov/sra/PRJNA799875 (accessed on 26 February 2022)). Two samples were excluded due to sequencing quality issues in the subsequent analysis. Soil fungi were divided into trophic modes and functional guilds by the FUNGuild database (http://funguild.org (accessed on 10 March 2022)). Confidence included “highly probable” and “probable”, but “possible” was excluded [56]. The analysis was performed using the R package ‘FUNGuildR’ (version 0.2.0.9) [57] in R version 4.1.1.

### 2.5. Statistical Analysis

The “soil total fungi” dataset contained the total OTUs detected by high-throughput sequencing. The soil fungal community dataset was divided into three subsets based on trophic modes: the “symbiotic fungi”, “saprophytic fungi”, and “pathogenic fungi” datasets (i.e., containing the OTUs of the trophic modes symbiotroph, saprotroph, and pathotroph, respectively.). The four fungal datasets (total, symbiotic, saprophytic, and pathogenic fungi) were analyzed separately. A Mantel test was used to check whether soil samples were independent or spatially autocorrelated [11]. The OTUs table was used for downstream community composition and diversity analysis. The relative abundance of soil fungal phyla levels, trophic modes, and functional guilds in each sample were calculated and ranked. Non-metric multi-dimensional scaling (NMDS) analysis based on the Bray–Curtis distance and analysis of similarities (ANOSIM) were carried out to examine soil fungal community composition dissimilarities among different vegetation types [58]. When exploring the relationship between altitude variation and soil fungal community composition dissimilarities, a generalized additive model was used within the *ordisurf* function in R package ‘vegan’ (version 2.6-2) [59]. The α-diversity was estimated using OTUs richness, the Simpson index, Pielou index, and Shannon–Wiener index [60]. To calculate β-diversity, the Jaccard index was used for the pairwise dissimilarity of species composition and the Bray–Curtis index was used for abundance-weighted dissimilarity. The β-diversity and its components (turnover and nestedness) were computed using the function *beta.pair* and *beta.pair.abund* in R package ‘betapart’ (version 1.5.4) [61]. Altitude distance (based on Euclidean distances) was computed using the function *vegdist* in R package ‘vegan’ (version 2.6-2).

To disentangle the relationship between soil fungal community composition and environmental variables, a distance-based redundancy analysis (db-RDA) and Monte Carlo permutation test (999 permutations) were performed. For the db-RDA analysis, the significance of a full model including all the explanatory variables was first tested and then the model was simplified by forward-model selection, using the function *ordiR2step* in R package ‘vegan’ (version 2.6-2). To evaluate the relationship between α-diversity and environmental variables, Pearson correlation analysis was used. The pairwise geographic distance (GEO) among sample plots was calculated using the R package ‘geosphere’ (version 1.5-14) [62], according to the geographic coordinates of each sample plot. Multiple regression on matrices (MRM) methods were used to examine the relative effects of geographic distance and dissimilarity in environmental variables (based on Euclidean distances) on β-diversity and its two components (based on the Bray–Curtis index). To exclude strong collinearity among environmental variables, environmental variables with high correlations were removed before the MRM test. To prevent the influence of data overfitting, two MRM tests were performed in R package ‘ecodist’ (version 2.0.9) [63]. The first MRM test was run to remove insignificant variables, then a second test was run with significant variables, and the model results for the second test were reported. All statistical analyses and plotting were conducted in R (version 4.1.1; http://www.r-project.org/ (accessed on 26 February 2022)).

## 3. Results

### 3.1. Soil Fungal Community Composition

A total of 2,776,959 sequences were obtained after quality control in 61 samples from seven vegetation types along altitude gradients. The number of sequences in each sample ranged from 5824 to 87,935, with an average of 45,524 ± 17,559. The sequences were clustered into 9036 OTUs and the rarefaction curve indicating the depth of the sequence is shown in Appendix A. The 9036 OTUs were assigned to 14 phyla, 52 classes, 130 orders, 282 families, and 653 genera. The soil fungal phylum Ascomycota was the most dominant (47.72% of total OTUs), followed by Basidiomycota (36.58%) and Mortierellomycota (12.14%); the remaining 11 phyla account jointly only for 3.56% (Figure 2a). The relative abundance of Ascomycota exhibited a hollow pattern along altitude (R^2^ = 0.61, *p* < 0.001), while Basidiomycota (R^2^ = 0.43, *p* < 0.001) and Mortierellomycota (R^2^ = 0.22, *p* < 0.001) were opposite diametrically, showing a hump-shaped pattern along altitude. The relative abundance of the top 10 orders and families in the phylum Ascomycota accounted for 70.33% and 41.01%, while those in Basidiomycota accounted for 91.20% and 72.58%, respectively. There was a variation in the relative abundance of orders and families among the vegetation types (Figure 2b,c). Some orders, such as Helotiales (17.21%), Hypocreales (12.82%), and Archaeorhizomycetales (8.73%), were the most dominant in the phylum Ascomycota (Figure 2b left). The soil fungal family Russulaceae (15.46%) was the most dominant in the phylum Basidiomycota, followed by Inocybaceae (13.81%) and Hygrophoraceae (10.77%) (Figure 2c right).

In total, 4503 OTUs (49.83% of the total OTUs) were assigned to seven trophic modes. After excluding the confidence level as a possible level, there were still 3457 OTUs (38.26% of the total OTUs). Among the seven trophic modes, the Symbiotroph accounted for 30.58% of assigned 3457 OTUs, followed by the Saprotroph–Symbiotroph (22.51%) and the Saprotroph (20.62%) (Figure 2d). The soil fungal functional guilds were dominated by Ectomycorrhizal, Ectomycorrhizal–Undefined Saprotroph, and Undefined Saprotroph, and the relative abundance of ectomycorrhizal fungi showed a hump-shaped pattern along altitude (Figure 2e).

Soil fungal (total and trophic modes) community composition differed significantly across vegetation types along altitude (ANOSIM: R = 0.347–0.673, *p* = 0.001) (Figure 3). There was a nonlinear relationship between altitude and community composition of soil total fungi, and altitude could explain 91% variation of community composition of soil total fungi (*p* < 0.001) (Figure 3a). For different trophic modes, altitude could explain more than 70% variation of their community composition (*p* < 0.001) (Figure 3b–d).

### 3.2. Soil Fungal Diversity

The mean OTUs richness of soil total fungi varied from 409 to 1230 (Figure 4a), and the Shannon–Wiener index of soil total fungi varied from 4.80 to 8.81 along altitude gradients (Figure 4b). The OTUs richness of soil fungi (total and trophic modes) had no obvious pattern along altitude gradients (*p* > 0.05). The Shannon–Weiner index (R^2^ = 0.231, *p* < 0.001), Simpson diversity index (R^2^ = 0.195, *p* = 0.002), and Pielou index (R^2^ = 0.260, *p* < 0.001) of soil total fungi showed a hollow pattern along altitude gradients. The α-diversity (apart from richness) of symbiotic fungi had a hollow pattern (*p* < 0.05), while saprophytic fungi and pathogenic fungi showed no obvious pattern along altitude gradients (Figure 4).

The turnover components, accounted for 93.45% by Bray–Curtis and 94.09% by Jaccard, contributed more to the β-diversity of soil total fungi than nestedness components (6.55%; 5.91%). Among different soil fungal trophic modes, turnover components accounted for 85.1−94.7% by Bray–Curtis and 93.3−96.5% by Jaccard of β-diversity (Table 2). The β-diversity and turnover components of soil fungi (total and trophic modes) increased linearly with increased altitude distance (*p* < 0.001). The slopes of turnover components of soil total fungi, saprophytic fungi, and pathogenic fungi with altitude distance were greater than that of β-diversity. The nestedness components of soil total fungi and saprophytic fungi decreased linearly with increased altitude distance (*p* < 0.001). The nestedness component of soil symbiotic fungi showed no significant response to altitude distance (*p* > 0.05).

### 3.3. Drivers of the Soil Fungal Community

Soil properties explained 34.60% of variation of soil total fungal community composition (F = 1.913, *p* = 0.001), in which the RDA1 and RDA2 axis explained 9.308% and 5.421%, respectively (Figure 5a). Soil properties explained 31.27%, 34.91%, and 32.35% of variations in community composition of soil symbiotic fungi, saprophytic fungi, and pathogenic fungi (*p* = 0.001), respectively (Figure 5b–d).

The community composition of soil fungi (total and trophic modes) was driven by EC, WFPS, pH, and soil nutrients, but not soil stoichiometry. The effects of WFPS, EC, and pH on the fungal community composition of different trophic modes were similar. The effects of TP, AP, and NO_3_^−^-N on the community composition of symbiotic fungi were greater than that of saprophytic fungi and pathogenic fungi. The effect of NH_4_^+^-N on the community composition of pathogenic fungi was greater than that of symbiotic fungi and saprophytic fungi (Table 3).

The soil pH was related positively to α-diversity of soil total fungi, and NO_3_^−^-N and TP also had similar relationships with α-diversity of soil total fungi (except when using the Pielou index). The soil pH, EC, and BD were related positively to α-diversity (using richness, the Shannon–Wiener index, and the Simpson index) of soil symbiotic fungi. For saprophytic fungi and pathogenic fungi, soil pH was correlated positively with richness (Figure 6).

Soil properties and geographic distance explained jointly the β-diversity of soil total fungi (27.57%) and saprophytic fungi (17.01%), while soil properties explained the β-diversity of symbiotic fungi (7.78%) and pathogenic fungi (6.59%), respectively. The turnover components of soil total fungi and saprophytic fungi were affected by soil properties and geographic distance, while those of symbiotic fungi and pathogenic fungi were affected only by soil properties. For the nestedness component of soil fungi (total and trophic modes), geographic distance did not have a significant effect (*p* > 0.05) and the explanatory power of soil properties was low (1.69−3.78%) (Table 4).

## 4. Discussion

### 4.1. Soil Fungal Community Composition

The community composition of soil fungi (total and trophic modes) changed significantly in different vegetation types, which reflected the response of the soil fungal community composition to altitude in the alpine–gorge region (Figure 2 and Figure 3). Ascomycota, Basidiomycota, and Mortierellomycota were the top three dominant phyla of soil fungal communities in the region (Figure 2a), which was similar to findings on the southeastern Qinghai–Tibetan Plateau [42]. Along the altitude gradient, the other 11 phyla of soil fungi in subalpine coniferous forests (E5) comprised the smallest proportion, accounting for only 1.44%, followed by subalpine broadleaf mixed forest (E3), accounting for 2.36% (Figure 2a). In mid-altitude, competition among soil fungi has resulted in an increasing proportion of the dominant phyla. The relative abundances of the top three dominant soil fungal phyla exhibited different altitude patterns (Figure 2a), indicating that there was niche differentiation among soil fungal taxa along altitude gradients [15]. According to the life history classification system based on functional traits, Grime’s C-S-R (competitor, stress tolerator, ruderal) framework [64], Ascomycota is stress-tolerant, which means that it is more resilient in harsh habitats (dry and cold) than other fungal taxa. Meanwhile, Basidiomycota (C-strategists) may have a stronger competitive advantage than other fungal taxa in better environmental conditions. This is consistent with previous findings on the opposite altitude distribution patterns of Ascomycota and Basidiomycota [15,16]. The relative abundance of soil symbiotic fungi and pathogenic fungi exhibited opposite altitude patterns (Figure 2d). This is consistent with previous knowledge that soil symbiotic fungi and pathogenic fungi have different suitable environmental conditions. Soil pathogenic fungi may proliferate under warmer and drier conditions, leading to increased plant disease [65]. Similarly, the relative abundance of soil pathogenic fungi in dry valley shrubs was significantly higher than in other vegetation types, implying that potential plant pathogens could cause root rot and other diseases in dry valley shrubs (Figure 2d). The soil fungal functional guilds were dominated by EcM along altitude, which was consistent with the findings of Zhao et al. [21]. In this study, the relative abundance of EcM fungi showed a hump-shaped pattern along altitude gradients (Figure 2e), and similar results were also reported in the study by Miyamoto et al. on Mount Fuji [66], mainly due to the fact that vegetation types at mid-altitude contained most of the EcM hosts *A. fargesii* var. *faxoniana*, *B. albosinensis* and *B. platyphylla* (Table 1).

### 4.2. Soil Fungal Diversity

There were no α-diversity maxima at mid-altitude in our study. On the contrary, a hollow pattern of α-diversity (except for richness) was displayed for soil total fungi along altitude (Figure 4). These results overturn our first hypothesis. Geml et al. [12] also found that the OTUs richness of soil total fungi lacked significant altitude patterns. The α-diversity patterns of soil total fungi were inconsistent with studies on the Kohala volcano of Hawaii [67] and Mount Kilimanjaro [24]. Along altitude, a monotonic decreasing pattern or no obvious pattern of soil total fungal α-diversity have been reported by several studies [24,68,69], yet a hollow pattern has rarely been reported. No general trends in α-diversity patterns were identified by summarizing changes in soil total fungi across global altitude gradients [20]. These results showed that the α-diversity pattern of soil total fungi along altitude gradients was inconclusive, suggesting different responses of soil total fungal α-diversity to local environmental variations along altitude [21]. The α-diversity of soil total fungi is related closely to plant distribution along altitude gradients [70], and different vegetation types may indirectly affect α-diversity by acting on local environmental factors [71]. Similar to the findings of previous studies [23,25], the α-diversity pattern of soil fungi varied by trophic modes along altitude gradients, with symbiotic fungi showing a hollow pattern while saprophytic fungi and pathogenic fungi showed no obvious patterns (Figure 4). For host-dependent symbiotic fungi, their α-diversity was more related closely to changes in vegetation types along altitude, given their mutually beneficial symbiotic lifestyles with their hosts [72]. Fewer hosts and root exudates at mid-altitude and more diverse root exudates in response to environmental stress at low and high altitudes may be responsible for the hollow patterns of symbiotic fungal α-diversity [73,74]. The α-diversity of saprophytic fungi remained stable along altitude gradients, which may be attributed to their greater adaptation to the external environment [75]. Additionally, α-diversity of symbiotic fungi and saprophytic fungi showed different patterns along altitude gradients (Figure 4), which may be due to the fact that symbiotic fungi (mainly EcM fungi) could act as biofilters for free-living saprophytic fungi [19]. EcM fungi and saprophytic fungi compete for the soil organic C and N, and EcM fungi could reduce saprophytic fungal abundance and diversity in the process [19]. In addition, limited shared resources and competitive exclusion were presumed to restrict the number of different fungal trophic modes coexisting in the same niche [23]. These different α-diversity altitude patterns may reflect competition for soil water and fertilizer and adaptation to the environment by soil fungi of different trophic modes [76].

The β-diversity of soil fungi (total and trophic modes) was derived mainly from the turnover processes rather than from the nestedness processes (Table 2), suggesting that soil fungal β-diversity is derived mainly from species turnover between different communities rather than species richness differences [77], and this is consistent with the results of previous studies [12,78]. These results supported our second hypothesis. The β-diversity of soil fungi (total and trophic mode) exhibited a significant altitude distance-decay pattern (Table 2), which is consistent with the results of previous studies [79,80], and provided further evidence for the strong effect of altitude distance on β-diversity in the region. Among different trophic modes, the slope of the relationship between altitude distance and turnover components of saprophytic fungi was the largest (Table 2), indicating that the species turnover of saprophytic fungi varied faster with altitude distance than that of symbiotic fungi and pathogenic fungi; this may be related to the fact that saprophytic fungi were more sensitive to environmental changes caused by altitude [23,71]. The relationship between the altitude distance and nestedness components varied by trophic modes (Table 2), which may be inseparable from the differences in relative fitness of different fungal trophic modes under environmental stress [81].

### 4.3. Drivers of the Soil Fungal Community

Generally, changes in soil fungal community composition along altitude gradients have been attributed to changes in both host distributions and environmental factors [82,83], with environmental factors playing a larger role than host species [84]. Our research showed clear evidence that soil WFPS, EC, pH, and nutrients jointly affected soil fungal community composition, and the effects of soil nutrients on soil fungal community composition varied by trophic modes (Table 3). This was consistent with previous findings that local environmental conditions determine soil fungal community composition [33], while the explanatory power of environmental variables varies by trophic modes [85]. Consistent with previous studies [86,87], soil moisture, EC, and pH had similar effects on the soil fungal community composition of different trophic modes (Table 3). Soil WFPS could potentially reflect the relative soil moisture gradient across sampling sites, and variations in soil WFPS could directly affect soil EC, which was related to soil fungal survival and activity [88]. The increase of soil EC increased extracellular osmolarity, and more salt-tolerant species may replace less salt-tolerant ones, thereby affecting the soil fungal community composition [89]. The findings of Jarvis et al. [83] confirmed our results that soil fungal community composition along altitude gradient was driven by soil moisture. Soil pH could directly affect soil fungal community composition by imposing physiological limitations on fungal growth and reproduction [90], while it could also indirectly affect fungal community composition by affecting other soil properties (e.g., soil nutrient availability) [15,87]. Due to different nutrient acquisition strategies, the relative importance of soil nutrients on soil fungal community composition varied among different trophic modes [91]. Soil symbiotic fungi form symbioses with host plants, which provide C to symbiotic fungi through photoassimilates (e.g., sucrose) and, in return, profit from mineral nutrients provided by soil symbiotic fungi [92]. Therefore, compared to soil mineral nutrients, SOC had less effect on the community composition of soil symbiotic fungi (Table 3). Soil mineral nutrients had the least effect on pathogenic fungi and the greatest effect on symbiotic fungi, which was similar to the results of previous studies [93]. The pathogenic fungi acquire all C and most mineral nutrients from their hosts and develop specializations on the host species [91,94], thus their community composition was influenced weakly by soil properties [95]. However, symbiotic fungi absorb soil nitrogen and phosphorus nutrients in exchange for C from the host, which results in their community composition being more influenced by soil nitrogen and phosphorus nutrients [96,97]. It should be noted that soil factors explained less than 35% of the variation in soil fungal community composition (Figure 5), and the unexplained variation in soil fungal community composition may be caused by unmeasured indicators such as soil temperature [22,36], plant factors [69,74], and mycophagous organisms [98,99,100].

Soil properties that vary with altitude also affect soil fungal α-diversity [48,101]. Soil pH was the most critical factor affecting soil fungal α-diversity in this study, which is consistent with the results of the latest research [12,28,40]. The α-diversity of soil total fungi increased with increasing soil pH (Figure 6), which is consistent with the results of a study on Mount Shegyla [28]. Soil pH may affect α-diversity of soil total fungi directly or indirectly by affecting other soil properties [39]. Moreover, soil pH was correlated positively with the richness of different soil fungal trophic modes (Figure 6). Similarly, Rincón et al. [102] also found a significantly positive relationship between soil pH and the richness of symbiotic fungi along altitude gradients. Soil TP was correlated positively with the richness of saprophytic fungi, but correlated negatively with the richness of symbiotic fungi (Figure 6), which was also confirmed by the results of Khalid et al. [103]. Saprophytic fungi depend on the soil substrate to absorb nutrients, so higher soil nutrients increased their richness [104]. Low phosphorus could stimulate increased phosphorus requirements in host plants, thereby increasing the richness of symbiotic fungi used for mycelial soil exploration [103]. The positive correlation between EC and richness of symbiotic fungi was greater than that of saprophytic fungi (Figure 6), which may be due to the fact that the plasticity of symbiotic fungi was higher than that of saprophytic fungi under salinity stress [105]. Symbiotic fungi were able to adjust richness by themselves and with the help of plants to counteract the detrimental effects of salinity, while saprophytic fungi could only adapt to high salt stress by adjusting their richness by themselves [105]. Inconsistent with the findings of Du et al. [106], there was a weak relationship between SOC and α-diversity of pathogenic fungi in our study, suggesting that other ecological factors may have a greater effect on α-diversity of pathogenic fungi [107].

Changes in β-diversity of soil fungi could be caused by environmental variations and geographic distance [20]. In this study, soil properties and geographic distance jointly drove the β-diversity of soil total fungi, while the effects of soil properties and geographic distance on soil fungal β-diversity varied by trophic modes (Table 4). The effect of geographic distance on soil fungal β-diversity varied by trophic modes, which may be related to their long-distance dispersal ability and lifestyle [108]. Unlike regional-scale studies [30], soil fungal β-diversity may be more directly driven by soil properties among environmental factors along altitude gradients. The β-diversity and its components of symbiotic fungi were affected by soil properties but not geographic distance (Table 4), which may be related to the limited dispersal ability of symbiotic fungi [109,110]. Similarly, Kivlin et al. [86] also found that soil moisture affects the turnover components of symbiotic fungi. Unlike symbiotic and pathogenic fungi, the β-diversity of saprophytic fungi was jointly affected by geographic distance and soil properties, which may be attributed to their free-living lifestyle and insensitivity to native plant species [92,111]. The β-diversity and turnover components of soil saprophytic fungi were driven mainly by changes in soil TN (Table 4), which was similar to the results of previous studies [84]. Owing to the lack of mutualistic interaction with plants, the β-diversity of soil saprophytic fungi was more affected by soil nutrient variation caused by geographic distance [76]. The nestedness components of soil fungi (total and trophic modes) were affected weakly by soil properties (1.69−3.78%) and were affected hardly at all by geographic distance (*p* > 0.05); this implies that nestedness components may be more affected by other environmental factors or processes (e.g., extinction–colonization dynamics) [78].

The soil fungi of facultative trophic modes in this paper have not been studied intensively, due to their complexity and uncertainty [112]. With the development of sequencing technology and database sharing [112,113], the classification of soil fungi will be more accurate, which will facilitate further research on soil fungi with facultative trophic modes [114]. Seasonal changes and their drivers in soil fungal community composition and diversity of different trophic modes along altitude gradients are also worth investigating [37,48]. Finally, our study was performed only along one mountainside. More studies containing four to five replicate altitude transects in the Qinghai–Tibet Plateau may elucidate more representative and general conclusions on the altitude variations of soil fungal community composition and diversity of different trophic modes.

## 5. Conclusions

In this study, we performed an integrated analysis of altitude variations and their drivers of soil fungal communities on the eastern Qinghai–Tibetan Plateau. Altitudinal changes in soil fungal communities of different trophic modes could act as a baseline for understanding the potential processes affecting soil fungal functional turnover and vegetation health. The results revealed in greater detail the role of soil factors in driving soil fungal community composition and diversity of different trophic modes along altitude gradients. Our study provided insights into the knowledge of soil fungal community variability of different trophic modes under potential environmental change in mountain ecosystems.

## Figures and Tables

**Figure 1 jof-08-00807-f001:**
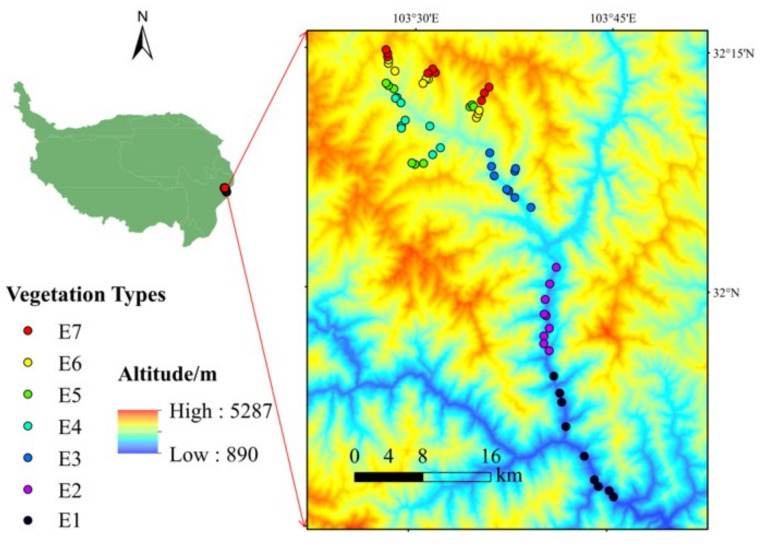
The sampling sites along altitude gradients of the study area on the eastern Qinghai–Tibetan Plateau.

**Figure 2 jof-08-00807-f002:**
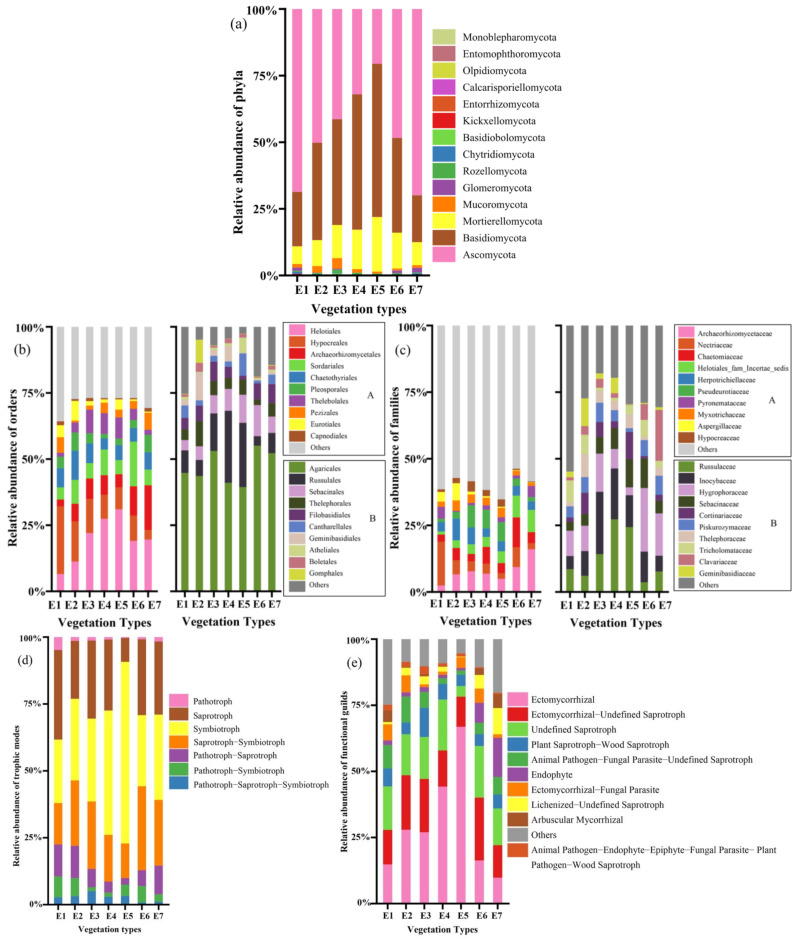
Relative abundance of the soil fungal phyla (**a**), the top 10 most abundant soil fungal orders (**b**) and families (**c**) belonging to dominant phyla Ascomycota and Basidiomycota, trophic modes (**d**), and top 10 most abundant functional guilds (**e**) among different altitude gradients. A and B in the figure represent soil fungal dominant phyla Ascomycota and Basidiomycota, respectively.

**Figure 3 jof-08-00807-f003:**
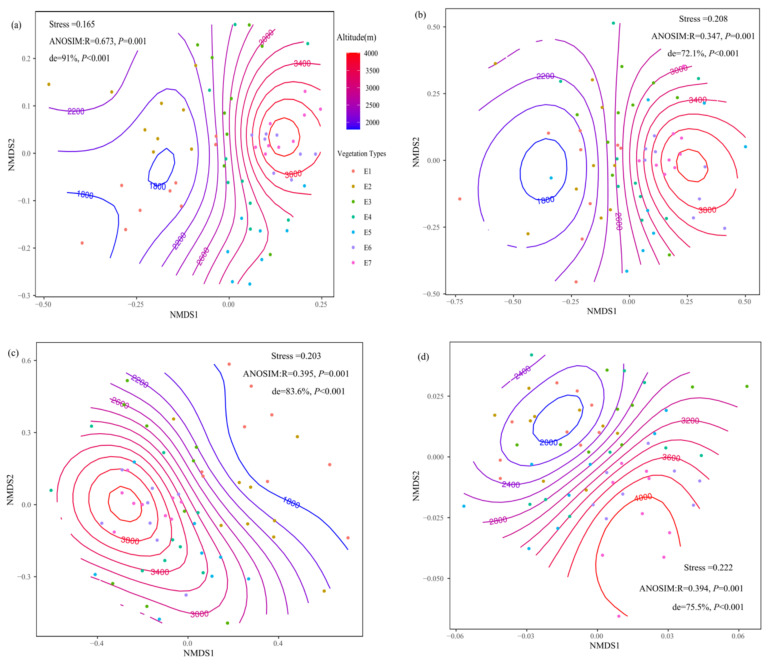
Non-metric multidimensional scaling (NMDS) and analysis of similarities (ANOSIM) represent community compositional dissimilarity of soil total fungi (**a**), symbiotic fungi (**b**), saprophytic fungi (**c**), and pathogenic fungi (**d**) among different altitude gradients. ‘de’ shows the deviance explained by the generalized additive model.

**Figure 4 jof-08-00807-f004:**
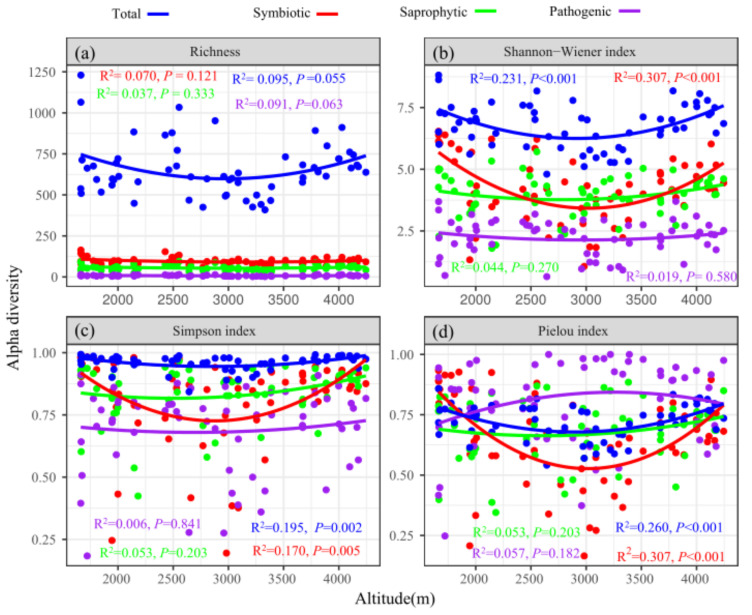
The α-diversity (richness (**a**), Shannon–Wiener index (**b**), Simpson index (**c**), Pielou index (**d**)) patterns of soil fungi along altitude gradients.

**Figure 5 jof-08-00807-f005:**
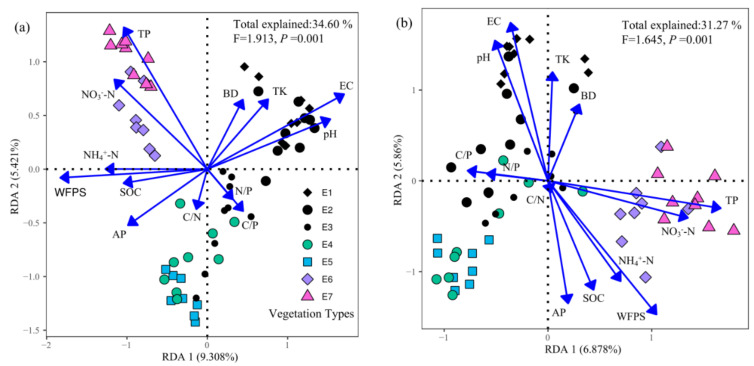
Distance-based redundancy analysis (db-RDA) demonstrated the effect of soil properties on community composition of soil total fungi (**a**), symbiotic fungi (**b**), saprophytic fungi (**c**), and pathogenic fungi (**d**).

**Figure 6 jof-08-00807-f006:**
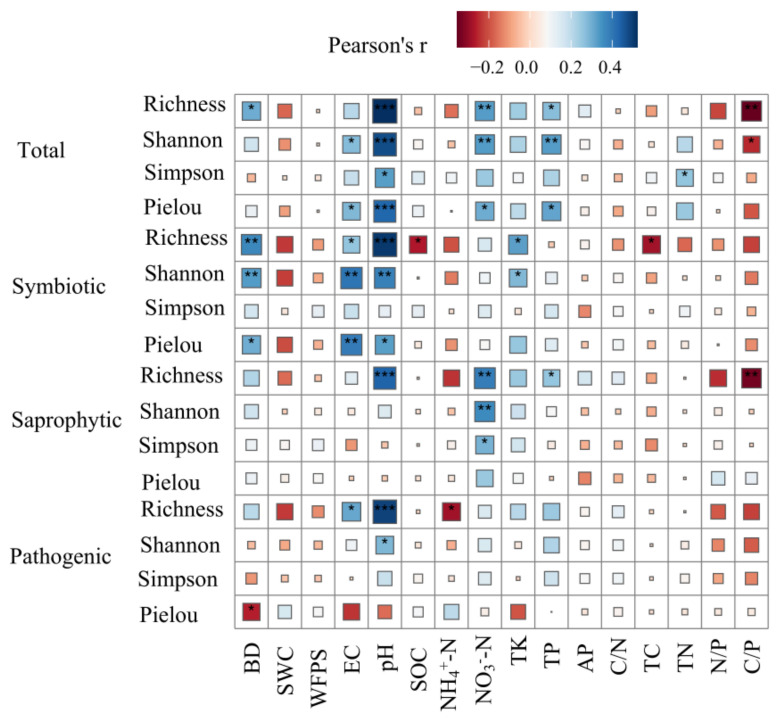
Pearson correlation analysis results of α-diversity of soil fungi and soil properties. Asterisks indicate significance level. *, *p* < 0.05; **, *p* < 0.01; ***, *p* < 0.001.

**Table 1 jof-08-00807-t001:** Description of sampling sites along altitude gradients in the alpine–gorge region. The listed plant species marked with * and # represent arbuscular mycorrhizal (AM) and ectomycorrhizal (EcM), respectively.

Abbreviation	Altitude Range (m)	Vegetation Types	Dominant Plant Species
E7	3998–4245	Alpine meadow	*Carex atrofusca, Polygonum viviparum, Anemone rivularis*
E6	3671–3893	Alpine shrub meadow	*Potentilla fruticose *, Sibiraea angustata, Berberis wilsoniae*
E5	3161–3516	Subalpine coniferous forest	*Abies fargesii* var. *faxoniana#*
E4	2805–3089	Subalpine coniferous-broadleaved mixed forest	*A. fargesii* var. *faxoniana #*, *Betula albosinensis #*
E3	2427–2769	Subalpine broadleaved mixed forest	*B. albosinensis* *#, B. platyphylla#, Acer davidii*
E2	1946–2182	Valley-mountain ecotone Broadleaved mixed forest	*Quercus baronii #, Cotinus coggygria *, Ostryopsis davidiana #*
E1	1663–1848	Dry valley shrub	*Campylotropis macrocarpa *, Bauhinia brachycarpa *, Sophora davidii **

**Table 2 jof-08-00807-t002:** General linear regression results between altitude distance and soil fungal β-diversity or its components as calculated by Bray–Curtis and Jaccard index.

Soil Fungi	Bray–Curtis Index				Jaccard Index			
	Component	Mean	Slope (km^−1^)	R^2^	Component	Mean	Slope (km^−1^)	R^2^
Total	β-diversity	0.850	0.0535	0.22 ***	β-diversity	0.750	0.0205	0.16 ***
	Turnover	0.794	0.0692	0.19 ***	Turnover	0.706	0.026	0.15 ***
	Nestedness	0.056	−0.0157	0.02 ***	Nestedness	0.044	−0.0055	<0.01 ***
Symbiotic	β-diversity	0.875	0.0205	0.02 ***	β-diversity	0.818	0.0110	0.03 ***
	Turnover	0.814	0.0191	0.01 ***	Turnover	0.786	0.0107	0.02 ***
	Nestedness	0.060	0.0014	<0.01 ns	Nestedness	0.032	0.00025	<0.01 ns
Saprophytic	β-diversity	0.775	0.0631	0.13 ***	β-diversity	0.798	0.0243	0.11 ***
	Turnover	0.660	0.1020	0.17 ***	Turnover	0.744	0.0318	0.09 ***
	Nestedness	0.115	−0.0388	0.06 ***	Nestedness	0.054	−0.00748	0.01 ***
Pathogenic	β-diversity	0.940	0.0332	0.06 ***	β-diversity	0.934	0.0299	0.08 ***
	Turnover	0.890	0.0548	0.07 ***	Turnover	0.901	0.0473	0.07 ***
	Nestedness	0.050	0.0217	0.03 ***	Nestedness	0.033	−0.0174	0.03 ***

Asterisks indicate significance level. ns, *p* > 0.05; ***, *p* < 0.001.

**Table 3 jof-08-00807-t003:** Results of Monte Carlo permutation tests for soil properties and soil fungal community composition.

Soil Properties	Total Fungi	Symbiotic Fungi	Saprophytic Fungi	Pathogenic Fungi
EC	0.799 ***	0.701 ***	0.713 ***	0.609 ***
WFPS	0.787 ***	0.742 ***	0.718 ***	0.757 ***
TP	0.618 ***	0.602 ***	0.464 ***	0.265 ***
pH	0.604 ***	0.595 ***	0.550 ***	0.591 ***
NO_3_^−^-N	0.454 ***	0.426 ***	0.407 ***	0.293 ***
NH_4_^+^-N	0.377 ***	0.390 ***	0.402 ***	0.513 ***
AP	0.294 ***	0.409 ***	0.295 ***	0.186 **
SOC	0.260 ***	0.368 ***	0.216 **	0.347 ***
TK	0.233 ***	0.308 **	0.170 **	0.261 ***
BD	0.141 **	0.163 **	0.079 ns	0.249 **
C/P	0.078 ns	0.128 *	0.046 ns	0.098 *
N/P	0.040 ns	0.074 ns	0.016 ns	0.095 ns
C/N	0.035 ns	0.004 ns	0.094 ns	0.014 ns

Asterisks indicate significance level. ns, *p* > 0.05; *, *p* < 0.05; **, *p* < 0.01; ***, *p* < 0.001.

**Table 4 jof-08-00807-t004:** Results of the MRM test for soil properties, geographic distance (GEO), and β-diversity.

Soil Fungi	Components	Fit Equation	R^2^
Total	β-diversity	0.0003EC + 0.0006C/P + 0.0008WFPS + 0.0066TN − 0.0007TC + 0.0009GEO	0.2757 ***
	Turnover	0.0011C/P + 0.0011WFPS + 0.0005EC + 0.0019GEO	0.2762 ***
	Nestedness	−0.0005WFPS	0.0279 ***
Symbiotic	β-diversity	0.0009C/P + 0.0006WFPS	0.0778 ***
	Turnover	0.0010C/P + 0.0006WFPS	0.0662 ***
	Nestedness	−0.0004SOC + 0.0023TK−0.0001WFPS	0.0240 *
Saprophytic	β-diversity	0.0121TN + 0.0005WFPS + 0.0025GEO	0.1701 ***
	Turnover	0.0009EC + 0.0154TN + 0.0006WFPS + 0.0031GEO	0.2470 ***
	Nestedness	0.0006C/P	0.0169 ns
Pathogenic	β-diversity	0.0048TN − 0.0005TC + 0.0008WFPS	0.0659 ***
	Turnover	0.0033TN + 0.0014WFPS	0.0569 ***
	Nestedness	0.0009SOC − 0.0041TN − 0.0006WFPS	0.0378 ***

Asterisks indicate significance level. ns, *p* > 0.05; *, *p* < 0.05; ***, *p* < 0.001.

## Data Availability

We have deposited the sequencing data in NCBI with accession number PRJNA799875.

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
