# Peer review of "Altitudinal Variation Influences Soil Fungal Community Composition and Diversity in Alpine–Gorge Region on the Eastern Qinghai–Tibetan Plateau"

_jof, 2022, doi:10.3390/jof8080807_

Round 1

Reviewer 1 Report

The manuscript is well-organised, very interesting and original. Few comments are provided to authors below:

L48: I would suggest "element biogeochemical cycles"

Materials and methods 

Please, provide versions for used R packages and more references to support applied methods. 

L384-385:"...suggesting…". Please, revise this part of the sentence as it is not clear. 

L400: what do you mean with "biofilters"? Please specify also in the text.

L475: "while…adjusting their richness". Please revise this sentence as it is not clear. 

Author Response

Response to Reviewer 1 Comments

Point 1: L48: I would suggest "element biogeochemical cycles"

Response 1: Thank you very much. We have revised them according to your suggestion. Please see L50 in the manuscript (clean) for further details.

Point 2: Materials and methods

Please, provide versions for used R packages and more references to support applied methods.

Response 2: Many thanks for your suggestions.  We have provided versions for used R packages and added corresponding references [51-63] to support applied methods. Please see L197-L254 in the manuscript (clean) for further details.

References

  1. Magoč, T.; Salzberg, S.L. FLASH: fast length adjustment of short reads to improve genome assemblies. Bioinformatics 2011, 27, 2957-2963, doi: 10.1093/bioinformatics/btr507.
  2. Edgar, R.C.; Haas, B.J.; Clemente, J.C.; Quince, C.; Knight, R. UCHIME improves sensitivity and speed of chimera detection. Bioinformatics 2011, 27, 2194-2200, doi: 10.1093/bioinformatics/btr381.
  3. Edgar, R.C. Search and clustering orders of magnitude faster than BLAST. Bioinformatics 2010, 26, 2460-2461, doi: 10.1093/bioinformatics/btq461.
  4. Kõljalg, U.; Nilsson, R.H.; Abarenkov, K.; Tedersoo, L.; Taylor, A.F.S.; Bahram, M.; Bates, S.T.; Bruns, T.D.; Bengtsson-Palme, J.; Callaghan, T.M.; et al. Towards a unified paradigm for sequence-based identification of fungi. Mol. Ecol. 2013, 22, 5271-5277, doi: 10.1111/mec.12481.
  5. Schloss, P.D.; Westcott, S.L.; Ryabin, T.; Hall, J.R.; Hartmann, M.; Hollister, E.B.; Lesniewski, R.A.; Oakley, B.B.; Parks, D.H.; Robinson, C.J.; et al. Introducing mothur: Open-Source, Platform-Independent, Community-Supported Software for Describing and Comparing Microbial Communities. Appl. Environ. Microb. 2009, 75, 7537-7541, doi: 10.1128/AEM.01541-09.
  6. Nguyen, N.H.; Song, Z.; Bates, S.T.; Branco, S.; Tedersoo, L.; Menke, J.; Schilling, J.S.; Kennedy, P.G. FUNGuild: An open annotation tool for parsing fungal community datasets by ecological guild. Fungal Ecol. 2016, 20, 241-248, doi: 10.1016/j.funeco.2015.06.006.
  7. Furneaux, B.; Song, Z. FUNGuildR: Look Up Guild Information for Fungi; R Package Version 0.2.0.9; Available online: https://github.com/brendanf/FUNGuildR/.
  8. Taguchi, Y.H.; Oono, Y. Relational patterns of gene expression via non-metric multidimensional scaling analysis. Bioinformatics 2005, 21, 730-740, doi: 10.1093/bioinformatics/bti067.
  9. Oksanen, J.; Blanchet, F.G.; Friendly, M.; Kindt, R.; Legendre, P.; McGlinn, D.; Minchin, P.R.; O’Hara, R.B.; Simpson, G.L.; Solymos, P.; et al. Vegan: Community Ecology Package; R Package Version 2.6-2; R Foundation for Statistical Computing: Vienna, Austria, 2022; Available online: https://cran.r-project.org/web/packages/ vegan/index.html.
  10. Willis, A.D. Rarefaction, Alpha Diversity, and Statistics. Front. Microbiol. 2019, 10, 2407, doi: 10.3389/fmicb.2019.02407.
  11. Baselga, A.; Orme, C.D.L. betapart: an R package for the study of beta diversity. Methods Ecol. Evol. 2012, 3, 808-812, doi: 10.1111/j.2041-210X.2012.00224.x.
  12. Hijmans, R.J.; Karney, C.; Williams, E.; Vennes, C. geosphere: Spherical Trigonometry; R Package Version 1.5-14; R Foundation for Statistical Computing: Vienna, Austria, 2022; Available online: https://cran.r-project.org/web/packages/geosphere/index.html.
  13. Goslee, S.; Urban, D. ecodist: Dissimilarity-Based Functions for Ecological Analysis; R Package Version 2.0-9; R Foundation for Statistical Computing: Vienna, Austria, 2022; Available online: https://cran.r-project.org/web/packages/ecodist/index.html.

Point 3: L384-385:"...suggesting…". Please, revise this part of the sentence as it is not clear.

“These results showed that α-diversity pattern of soil total fungi along elevation gradients was inconclusive, suggesting that different responses of soil total fungi to local environmental variations along elevation.”

Response 3: Many thanks for your suggestion. We have revised the sentence as “These results showed that α-diversity pattern of soil total fungi along elevation gradients was inconclusive, suggesting that different responses of soil total fungal α-diversity to local environmental variations along elevation [21].”. Please see L408-L411 in the manuscript (clean) for further details.

Reference

  1. Zhao, Y.; Zhou, Y.; Jia, X.; Han, L.; Liu, L.; Ren, K.; Ye, X.; Qu, Z.; Pei, Y. Soil characteristics and microbial community structure on along elevation gradient in a Pinus armandii forest of the Qinling Mountains, China. Forest Ecol. Manag. 2022, 503, 119793, doi: 10.1016/j.foreco.2021.119793.

Point 4: L400: what do you mean with "biofilters"? Please specify also in the text.

Response 4: Thank you for your comment. "biofilters" mean that “EcM fungi and saprotrophic fungi compete for the soil organic C and N and EcM fungi could reduce saprotrophic fungal abundance and diversity in the process [19].” It has been added in L426-L428 in the manuscript (clean).

Reference

  1. Looby, C.I.; Martin, P.H. Diversity and function of soil microbes on montane gradients: the state of knowledge in a changing world. FEMS Microbiol. Ecol. 2020, 96, a122, doi: 10.1093/femsec/fiaa122.

Point 5: L475: "while…adjusting their richness". Please revise this sentence as it is not clear.

Response 5: Many thanks for your suggestion. We have revised the sentence as “Symbiotrophic fungi were able to adjust richness by themselves and with the help of plants to counteract the detrimental effects of salinity, while saprotrophic fungi could only adapt to high salt stress by adjusting their richness by themselves [105].” Please see L503-L506 in the manuscript (clean) for further details.

Reference

  1. Lin, L.; Jing, X.; Lucas-Borja, M.E.; Shen, C.; Wang, Y.; Feng, W. Rare taxa drive the response of soil fungal guilds to soil salinization in the Taklamakan desert. Front. Microbiol. 2022, 13, 862245, doi: 10.3389/fmicb.2022.862245.

Reviewer 2 Report

The topic addressed by the authors is interesting, statistical analyses are correct and clearly presented, and the discussion is well described. However, some elements of the manuscript should be improved, in order to increase the importance of the study.

1. Taxonomic identity of identified fungal OTU.
Authors have found 9 036 OTUs in this study, but the taxonomic identity of these OTUs is limited to the phylum level. That is not correct.

Especially Ascomycota and Basidiomycota, which are represented by 80% OTUs identified by the Authors, are necessary to be presented using a more detailed approach. The order or family level of taxonomic identity of OTUs is more appropriate.

So please, add Figure 2C with the order-level and family-level identity of OTUs out of the two most important and most abundant phyla, it is Ascomycota and Basidiomycota. 

For example, look at Figure 4 in this manuscript:  https://www.mdpi.com/1999-4907/12/3/353. Although I think, the bar chart (as used by the Authors) is more readable than the pie chart (used in Fig.4 in the aforementioned paper), the taxonomic identity of OTUs in this paper is appropriate.

2. The trophic guilds.
Figure S2 should be placed in the main body of the manuscript. I suggest merging the data for Trophic modes (i.e. symbiotroph, saprotroph etc) and for Fungal guilds (presented in Fig.S2). For example, the merged trophic modes and guilds are presented in Fig.1 in this paper: https://doi.org/10.1016/j.funeco.2015.06.006 .

3. Study area and research design
In Figure 1, the Authors have shown the sampling area along the 60km gradient. However, the vegetation types E1, E2 and E3 are grouped in distinct 10km long transects (E1 and E2 are marked by similar colours, I suggest to modified the colour of one of these two). On the contrary, vegetation types E4-E7 are located close to each other.

I suggest the Authors use the Mantel test, in order to check the autocorrelations among soil samples within the vegetation types and between them. The test should be prepared in order to obtain certainty, that pseudoreplication is avoided.

Moreover, I suggest the Authors modify the sampling design, obviously for the purpose of other studies. In the sampling area presented in Figure 1, we can see, that although the sampling area is quite huge (60km transect), it covers a single mountainside. Therefore, the fungal propagules can be easily transported from the higher elevations (E4-E7 vegetation types) down to lower elevations (E1-E3 vegetation types).

That form of sampling design is not sufficient to draw general conclusions, because technically, it is the kind of case study. 

For further studies, I strongly recommend the Authors use the replication of the transects. I consider, that almost 4-5 mountainsides separate from each other (for example by a 50km distance) could be adequate. Then the results would be more conclusive and representative to the ecosystems of the Qinghai-Tibetan Plateau.  I hope, the Authors consider this suggestion carefully.

Author Response

Response to Reviewer 2 Comments

(Since it is inconvenient to show figures and tables here, please see the attachment for detailed reply information.)

Point 1: 1. Taxonomic identity of identified fungal OTU. Authors have found 9 036 OTUs in this study, but the taxonomic identity of these OTUs is limited to the phylum level. That is not correct.

Especially Ascomycota and Basidiomycota, which are represented by 80% OTUs identified by the Authors, are necessary to be presented using a more detailed approach. The order or family level of taxonomic identity of OTUs is more appropriate.

So please, add Figure 2C with the order-level and family-level identity of OTUs out of the two most important and most abundant phyla, it is Ascomycota and Basidiomycota.

For example, look at Figure 4 in this manuscript:  https://www.mdpi.com/1999-4907/12/3/353. Although I think, the bar chart (as used by the Authors) is more readable than the pie chart (used in Fig.4 in the aforementioned paper), the taxonomic identity of OTUs in this paper is appropriate.

Response 1: Many thanks for your comments and suggestions. Since there are seven vegetation types along elevation gradients in this paper, the two dominant phyla contain many orders and families (Ascomycota: 63 orders and 159 families; Basidiomycota: 43 orders and 101 families). Although we selected the relative abundance of top 10 orders and families to display by donut chart, data in inner donut chart was not displayed clearly. Therefore, the relative abundance of the top 10 orders and families belonging to the dominant phyla Ascomycota and Basidiomycota was shown by a bar chart respectively. We have added some descriptions of the dominant orders and families in phyla Ascomycota and Basidiomycota in the “3.1. Soil fungal community composition” section of the manuscript (clean) according to Figure 2b and Figure 2c. Please see L268-L275 in the manuscript (clean) for further details.

Figure 2. Relative abundance of the top 10 most abundant soil fungal orders (b) and families (c) in dominant phyla Ascomycota and Basidiomycota among different elevation gradients. A and B in figure represent soil fungal dominant phyla Ascomycota and Basidiomycota, respectively.

Point 2: 2. The trophic guilds.

Figure S2 should be placed in the main body of the manuscript. I suggest merging the data for Trophic modes (i.e.symbiotroph, saprotroph etc) and for Fungal guilds (presented in Fig.S2). For example, the merged trophic modes and guilds are presented in Fig.1 in this paper: https://doi.org/10.1016/j.funeco.2015.06.006 .

Response 2: Many thanks for your comments and suggestions. We have moved Figure S2 into the main body of the manuscript (Figure 2e). It is not easy to merge trophic modes and functional guilds due to the existence of facultative trophic modes and functional guilds. Meanwhile, the elevation response patterns of relative abundance of the different soil fungal trophic modes were shown in Figure 2d, and this useful information would be lost if Figures 2d and Figures 2e were merged. Considering the completeness and intuitiveness of the data presentation, we keep two figures (trophic modes and functional guilds) in the manuscript.

Figure 2. Relative abundance of the soil fungal trophic modes (d) and top 10 most abundant functional guilds (e) among different elevation gradients.

Point 3: 3. Study area and research design

In Figure 1, the Authors have shown the sampling area along the 60km gradient. However, the vegetation types E1, E2 and E3 are grouped in distinct 10km long transects (E1 and E2 are marked by similar colours, I suggest to modified the colour of one of these two). On the contrary, vegetation types E4-E7 are located close to each other.

Response 3: Many thanks for your comments and suggestions. We modified the color of E1 in order to distinguish the colors of E1 and E2. Please see Figure 1 (L135) in the manuscript (clean) for further details.

Figure 1. The sampling sites along elevation gradients of the study area on the eastern Qinghai-Tibetan Plateau.

I suggest the Authors use the Mantel test, in order to check the autocorrelations among soil samples within the vegetation types and between them. The test should be prepared in order to obtain certainty, that pseudoreplication is avoided.

Response: Many thanks for your comments and suggestions. We have performed mantle tests on soil properties within the same vegetation types and between different vegetation types, as shown in the table below. There was no spatial autocorrelation among soil samples within the vegetation types (P > 0.05) (Table R1). The soil properties among different vegetation types were not correlated significantly (P > 0.05) (Table R2). Therefore, the problem of pseudoreplication could be excluded. In addition, “A Mantel test was used to check whether soil samples were independent or spatial autocorrelation [11].” has been added in L219-L220 in the manuscript (clean). Please see that in the manuscript (clean) for further details.

Reference

  1. Peay, K.G.; Baraloto, C.; Fine, P.V. Strong coupling of plant and fungal community structure across western Amazonian rainforests. ISME J 2013, 7, 1852-1861, doi: 10.1038/ismej.2013.66.

Table R1. Mantle test (999 permutations) results for soil properties and sampling spatial locations. ns indicated P > 0.05.

Mantel statistic r

Significance

E1

0.220

0.078ns

E2

-0.061

0.591 ns

E3

0.003

0.487 ns

E4

0.294

0.073ns

E5

0.006

0.451 ns

E6

0.052

0.329 ns

E7

0.101

0.260 ns

Table R2. Mantle test (999 permutations) results of soil properties between different vegetation types. Asterisks indicated significant level. ns, P > 0.05; ***, P < 0.001.

E1

E2

E3

E4

E5

E6

E7

E1

1***

0.053 ns

-0.243 ns

-0.139 ns

-0.037 ns

0.154 ns

0.322 ns

E2

0.053 ns

1***

0.052 ns

-0.128 ns

-0.017 ns

0.150 ns

-0.224 ns

E3

-0.243 ns

0.052 ns

1***

-0.220 ns

-0.070 ns

-0.226 ns

-0.039 ns

E4

-0.139 ns

-0.128 ns

-0.220 ns

1***

-0.051 ns

-0.011 ns

0.190 ns

E5

-0.037 ns

-0.017 ns

-0.070 ns

-0.051 ns

1***

0.268 ns

0.107 ns

E6

0.154 ns

0.150 ns

-0.226 ns

-0.011 ns

0.268 ns

1***

-0.135 ns

E7

0.322 ns

-0.224 ns

-0.039 ns

0.190 ns

0.107 ns

-0.135 ns

1***

Moreover, I suggest the Authors modify the sampling design, obviously for the purpose of other studies. In the sampling area presented in Figure 1, we can see, that although the sampling area is quite huge (60km transect), it covers a single mountainside. Therefore, the fungal propagules can be easily transported from the higher elevations (E4-E7 vegetation types) down to lower elevations (E1-E3 vegetation types).

Response: Many thanks for your comments and suggestions. The sampling design has been modified carefully and more details have been added. Please see L139-L156 in the manuscript (clean) for further details. As you said, the fungal propagules may be transferred from the higher elevations (E4-E7 vegetation types) to lower elevations (E1-E3 vegetation types). However, the effect of fungal propagules on mineral soils may be less due to the barrier effect of litter and humus [1,2].

References

  1. Koizumi, T.; Hattori, M.; Nara, K. Ectomycorrhizal fungal communities in alpine relict forests of Pinus pumila on Mt. Norikura, Japan. Mycorrhiza 2018, 28, 129-145, doi: 10.1007/s00572-017-0817-5.
  2. Santalahti, M.; Sun, H.; Jumpponen, A.; Pennanen, T.; Heinonsalo, J. Vertical and seasonal dynamics of fungal communities in boreal Scots pine forest soil. FEMS Microbiol. Ecol. 2016, 92, w170, doi: 10.1093/femsec/fiw170.

That form of sampling design is not sufficient to draw general conclusions, because technically, it is the kind of case study.

For further studies, I strongly recommend the Authors use the replication of the transects. I consider, that almost 4-5 mountainsides separate from each other (for example by a 50km distance) could be adequate. Then the results would be more conclusive and representative to the ecosystems of the Qinghai-Tibetan Plateau.  I hope, the Authors consider this suggestion carefully.

Response: Many thanks for your comments and suggestions. Indeed, this paper is just a case study and draws a case conclusion. We fully agree with you that almost 4-5 mountainsides separate from each other (for example by a 50 km distance) could be adequate in order to draw more conclusive and representative results to the ecosystems of the Qinghai-Tibetan Plateau. It will be conducted in our future study. Additionally, we have added more details for further studies as you suggested in the Discussion (L537-L540) of the manuscript (clean). Please see that in the manuscript (clean) for further details.

Reviewer 3 Report

This manuscript presents a study on the variation in soil fungal community composition and diversity in seven vegetation types along an altitudinal gradient in Alpine-Gorge Region on the Eastern Qinghai-Tibetan Plateau. The results of the study indicate that the soil fungal community along the altitudinal gradient was dominated by fungi belonging to Ascomycota followed by Basidiomycota, and Mortierellomycota. Although soil pH was found to be the dominant soil factor influencing the diversity of soil fungi along the altitudinal gradient, the influence of other soil factors tended to vary with the nutritional modes of the fungi. From the results, it was concluded that the soil fungal community composition and diversity as influenced by soil properties differed with nutritional modes of the fungi along the elevation gradients in the alpine-gorge region. These conclusions are not exciting as the influence of soil factors on the distribution of fungal communities along the altitudinal gradient is well reported in the literature as the several studies cited by the author in the manuscript. In addition, several considerations need attention. As the comments and changes are numerous to list here, I marked many of my comments and changes directly in the annotated manuscript. 

        The results of the study should be interpreted with caution as the results are based on a single sampling during a specific part of a year. Seasonal variation in soil fungal diversity and abundance is a well-known response. In addition, other environmental factors like the stages of plant growth, climatic conditions, etc., can also substantially influence the diversity of soil fungi.

        Provide more details on the sampling. From the methods, it is not clear if the soil samples in all the vegetations were collected simultaneously or on different dates of the specified month. Moreover, provide details on the methods used to determine the sampling area size and their numbers. Were nested quadrats and species-area curves used to fix the sampling area size and sampling number?

        Some of the results presented need to be verified. For instance, Lines 282–284 state “The alpha-diversity (apart from richness) of symbiotrophic fungi had a hollow pattern (< 0.05), while saprotrophic fungi and pathotrophic fungi showed no obvious pattern along elevation gradients” while the data presented in Figure 4 shows that the Shannon-Wiener index, Simpson index and Pielou index for saprophytic fungi were significant. 

        Similarly, the redundancy analysis (Figure 5) explains only <35% of the variation that is accounted for by soil factors. This clearly shows that the majority of the variation (>65%) remains unexplained for most of the fungal groups. This needs a critical discussion in the relevant section.

        Several important aspects were not considered in the discussion. For example, there are very few obligate symbiotic and pathogenic fungi in any ecosystem as many of these fungi may also adopt a saprophytic mode of nutrition in the absence of specific host species. For example, Fusarium species that cause diseases in several plant species also possess isolates that are shown to improve plant growth. Therefore, exercise caution while categorizing fungi based on their nutritional modes.

        Another important aspect that was neglected in the present study is the role of mycophagous soil fauna which are known to affect soil fungal diversity. For example, see https://doi.org/10.1016/j.funeco.2021.101046; https://doi.org/10.11646/zootaxa.5114.1.3; https://doi.org/10.1038/s41598-019-50462-z. Changes in the populations of these mycophagous organisms could also affect the diversity of soil fungi.

Author Response

Response to Reviewer 3 Comments

 (Since it is inconvenient to show figures and tables here, please see the attachment for detailed reply information.)

Point 1: The results of the study should be interpreted with caution as the results are based on a single sampling during a specific part of a year. Seasonal variation in soil fungal diversity and abundance is a well-known response. In addition, other environmental factors like the stages of plant growth, climatic conditions, etc., can also substantially influence the diversity of soil fungi.

Response 1: Many thanks for your comments and suggestions. In this study, we just focus on changes of soil fungal communities along elevation gradients in the same growing season. Yes, seasonal variation in soil fungal diversity and abundance is a well-known response and it will be studied together with other environmental factors like the stages of plant growth, climatic conditions, etc. in the future according to your comments. Additionally, we have added a critical discussion (L482-L485) and more details for further studies (L535-L537) as you suggested in the Discussion of the manuscript (clean). Please see that in the manuscript (clean) for further details.

Point 2: Provide more details on the sampling. From the methods, it is not clear if the soil samples in all the vegetations were collected simultaneously or on different dates of the specified month. Moreover, provide details on the methods used to determine the sampling area size and their numbers. Were nested quadrats and species-area curves used to fix the sampling area size and sampling number?

Response 2: Thank you for your comments and suggestions. The soil samples in all the vegetations were collected from low elevation to high elevation within a week (different dates for different vegetations). We have provided more details on sampling times. Please see L151-L152 in the manuscript (clean) for further details. “In July 2020” has been revised as “In late July 2020”. Please see L139 in the manuscript (clean) for further details. The area size and numbers of sampling plots were determined through field investigation and literature review [9,28,48]. By reviewing previous studies on changes in microbial diversity along elevation gradients on the Qinghai-Tibet Plateau, it was found that the plot area was 20 m×20 m for forests, and 10 m×10 m for shrubs and meadows. We have added more details on the sampling in “2.2. Plot setup and soil sampling” section (L139-L156) of the manuscript (clean). Please see that in the manuscript (clean) for further details.

References

  1. Duan, Y.; Lian, J.; Wang, L.; Wang, X.; Luo, Y.; Wang, W.; Wu, F.; Zhao, J.; Ding, Y.; Ma, J.; et al. Variation in soil microbial communities along an elevational gradient in alpine meadows of the Qilian Mountains, China. Front. Microbiol. 2021, 12, 684386, doi: 10.3389/fmicb.2021.684386.
  2. Wang, J.; Zheng, Y.; Hu, H.; Zhang, L.; Li, J.; He, J. Soil pH determines the alpha diversity but not beta diversity of soil fungal community along altitude in a typical Tibetan forest ecosystem. J. Soil. Sediment. 2015, 15, 1224-1232, doi: 10.1007/s11368-015-1070-1.
  3. Gong, S.; Feng, B.; Jian, S.P.; Wang, G.S.; Ge, Z.W.; Yang, Z.L. Elevation matters more than season in shaping the heterogeneity of soil and root associated ectomycorrhizal fungal community. Microbiol. Spectr. 2022, 10, e195021, doi: 10.1128/spectrum.01950-21.

Point 3: Some of the results presented need to be verified. For instance, Lines 282–284 state “The alpha-diversity (apart from richness) of symbiotrophic fungi had a hollow pattern (P < 0.05), while saprotrophic fungi and pathotrophic fungi showed no obvious pattern along elevation gradients” while the data presented in Figure 4 shows that the Shannon-Wiener index, Simpson index and Pielou index for saprophytic fungi were significant.

Response 3: Many thanks for your suggestions. We have checked carefully Figure 4 and found no data presentation errors. Meanwhile, we have adjusted the spacing of the legend and modified the legend text (replaced “symbiotrophic, saprotrophic, pathotrophic”) with “symbiotic, saprophytic, pathogenic”) to let it more clear. Please see Figure 4 (L311) in the manuscript (clean) for further details.

Figure 4. The α-diversity patterns of soil fungi along elevation gradients.

Point 4: Similarly, the redundancy analysis (Figure 5) explains only <35% of the variation that is accounted for by soil factors. This clearly shows that the majority of the variation (>65%) remains unexplained for most of the fungal groups. This needs a critical discussion in the relevant section.

Response 4: Thank you for your comments and suggestions. We have added a critical discussion as you suggested in the “4.3. The drivers of soil fungal community” section of the manuscript (clean). Please see L482-L485 in the manuscript (clean) for further details.

Point 5: Several important aspects were not considered in the discussion. For example, there are very few obligate symbiotic and pathogenic fungi in any ecosystem as many of these fungi may also adopt a saprophytic mode of nutrition in the absence of specific host species. For example, Fusarium species that cause diseases in several plant species also possess isolates that are shown to improve plant growth. Therefore, exercise caution while categorizing fungi based on their nutritional modes.

Response 5: Thank you for your valuable comments. Soil fungi are abundant and interesting, and may produce facultative trophic modes in response to different environmental changes. In this study, soil fungi were divided into seven trophic modes (including four facultative trophic modes) by the FUNGuild database (http://funguild.org) [56], which is a popular and used commonly database with scientific and credible classification results [5,17,23]. The soil fungi of facultative trophic modes have not been studied in this study and they will be paid attention in our future study. Additionally, we have added more details for further studies as you suggested in the Discussion (L531-L535) of the manuscript (clean). Please see that in the manuscript (clean) for further details.

References

  1. Delgado-Baquerizo, M.; Guerra, C.A.; Cano-Díaz, C.; Egidi, E.; Wang, J.; Eisenhauer, N.; Singh, B.K.; Maestre, F.T. The proportion of soil-borne pathogens increases with warming at the global scale. Nat. Clim. Change 2020, 10, 550-554, doi: 10.1038/s41558-020-0759-3.
  2. Veach, A.M.; Stokes, C.E.; Knoepp, J.; Jumpponen, A.; Baird, R. Fungal communities and functional guilds shift along an elevational gradient in the southern Appalachian Mountains. Microb. Ecol. 2018, 76, 156-168, doi: 10.1007/s00248-017-1116-6.
  3. Yang, T.; Tedersoo, L.; Fu, X.; Zhao, C.; Liu, X.; Gao, G.; Cheng, L.; Adams, J.M.; Chu, H. Saprotrophic fungal diversity predicts ectomycorrhizal fungal diversity along the timberline in the framework of island biogeography theory. ISME commun. 2021, 1, 15, doi: 10.1038/s43705-021-00015-1.
  4. Nguyen, N.H.; Song, Z.; Bates, S.T.; Branco, S.; Tedersoo, L.; Menke, J.; Schilling, J.S.; Kennedy, P.G. FUNGuild: An open annotation tool for parsing fungal community datasets by ecological guild. Fungal Ecol. 2016, 20, 241-248, doi: 10.1016/j.funeco.2015.06.006.

Point 6: Another important aspect that was neglected in the present study is the role of mycophagous soil fauna which are known to affect soil fungal diversity. For example, see https://doi.org/10.1016/j.funeco.2021.101046; https://doi.org/10.11646/zootaxa.5114.1.3; https://doi.org/10.1038/s41598-019-50462-z. Changes in the populations of these mycophagous organisms could also affect the diversity of soil fungi.

Response 6: Many thanks for your professional comment. The role of mycophagous soil fauna which are known to affect soil fungal diversity has not been discussed in this study. Effects of mycophagous soil fauna on soil fungi will be paid more attention in our future studies. Additionally, we have added a critical discussion (L482-L485) as you suggested in the Discussion of the manuscript (clean). Please see that in the manuscript (clean) for further details.

Point 7: L2:modify this sentence: “Elevation Variability and Its………”.

Response 7: Thank you for your professional suggestion.  We have modified this sentence to “Elevation Variations and their.........”. Please see L2 in the manuscript (clean) for further details.

Point 8: L20: modify this sentence:

“The understanding of the elevation variability and its drivers of soil fungal community are still relatively unclear.”.

Response 8: Many thanks for your suggestion. We have modified this sentence to “The understanding of the elevation variations and their drivers of soil fungal community composition and diversity remains relatively unclear.”. Please see L19-L21 in the manuscript (clean) for further details.

Point 9: L24: Specify what these vegetation types were.

Response 9: Many thanks for your suggestions. We have supplemented with specific information for seven vegetation types. “seven vegetation types (dry valley shrub, valley-mountain ecotone broadleaved mixed forest, subalpine broadleaved mixed forest, subalpine coniferous-broadleaved mixed forest, subalpine coniferous forest, alpine shrub meadow, alpine meadow)” has been added in L24-L26 in the manuscript (clean). Please see L24-L26 in the manuscript (clean) for further details.

Point 10: L30-L31: symbiotrophic fungi, saprotrophic fungi, and pathotrophic fungi:

Use more commonly used terms like the symbiotic fungi, pathogenic fungi and saprophytic fungi.

Moreover, symbiotic is more generalized term as it may indicate mutualistic as well as pathogenic associations.

Response 10: Many thanks for your professional suggestions. We have replaced these words (symbiotrophic fungi, saprotrophic fungi, and pathotrophic fungi) with more common words (symbiotic fungi, saprophytic fungi, and pathogenic fungi), and revised them throughout the manuscript. Please see that in the manuscript (clean) for further details.

Point 11: L38: modify this sentence:“This study deepened our knowledge of the elevation variability and its drivers of soil fungal community composition and diversity and confirmed that the effects of soil properties on soil fungal community composition and diversity varied by trophic modes along elevation gradients in the alpine-gorge region.”

Response 11: Thank you for your suggestion. We have modified this sentence to “This study deepens our knowledge regarding the elevation variations and their drivers of soil fungal community composition and diversity and confirmed that the effects of soil properties on soil fungal community composition and diversity varied by trophic modes along elevation gradients in the alpine-gorge region.”. Please see L40-L43 in the manuscript (clean) for further details.

Point 12: L89: soil C/N: Explain the abbreviation before it is used.

Response 12: Thank you for your comment. “soil C/N” has been revised as “soil carbon to nitrogen ratio (C/N)”. Please see L91-L92 in the manuscript (clean) for further details.

Point 13: L151: Table 1: Dominant plant species: Please indicate the method used for assessing dominance.

Response 13: Many thanks for your comment. We have supplemented the method for determining the dominant plant species. “Plant community surveys were conducted in each plot, and importance values were calculated to determine plant dominant species [49].” has been added in L145-L146 in the manuscript (clean). Please see L145-L146 in the manuscript (clean) for further details.

Reference

  1. Li, S.; Huang, X.; Shen, J.; Xu, F.; Su, J. Effects of plant diversity and soil properties on soil fungal community structure with secondary succession in the Pinus yunnanensis forest. Geoderma 2020, 379, 114646, doi: 10.1016/j.geoderma.2020.114646.

Point 14: L138: There were nine plots per vegetation type?

Response 14: Thank you for your comment. Yes, there were nine plots per vegetation type. We have supplemented the plot numbers for each vegetation type in the manuscript (2.2. Plot setup and soil sampling). Please see L143 in the manuscript (clean) for further details.

Point 15: L145: Mention the quantity of the soil samples collected.

Response 15: Thank you very much for your suggestion. We have supplemented the quantity of the soil samples collected in the manuscript (2.2. Plot setup and soil sampling). Please see L151-L152 in the manuscript (clean) for further details.

Point 16: L154: “48 hours in an oven at 105 °C”: This is not a reliable parameter as it could exhibit great variations with time and space.

Response 16: Many thanks for your comment. We have modified “48 hours in an oven at 105 °C” to “dried to constant weight at 105 °C”. Please see L162 in the manuscript (clean) for further details.

Point 17: L174: What was the quantity of soil used for DNA extraction?

Response 17: Thank you for your comment. We have supplemented the quantity of soil samples used for DNA extraction in the manuscript (2.4. DNA extraction, MiSeq sequencing, and bioinformatics). Please see L182-L183 in the manuscript (clean) for further details.

Point 18: L207-208: modify this sentence.

Response 18: Thank you for your suggestion. We replaced the original words (symbiotrophic, saprotrophic, and pathotrophic fungi) with more commonly used terms like symbiotic, saprophytic, and pathogenic fungi. Please see L216-L219 in the manuscript (clean) for further details.

Point 19: L213, L218: Cite suitable references for the methods used.

Response 19: Thank you very much for your comments. We have added appropriate references [58,60] in the “2.5. Statistical analysis” section (L225 and L229) of the manuscript (clean). Please see L225 and L229 in the manuscript (clean) for further details.

References

  1. Taguchi, Y.H.; Oono, Y. Relational patterns of gene expression via non-metric multidimensional scaling analysis. Bioinformatics 2005, 21, 730-740, doi: 10.1093/bioinformatics/bti067.
  2. Willis, A.D. Rarefaction, Alpha Diversity, and Statistics. Front. Microbiol. 2019, 10, 2407, doi: 10.3389/fmicb.2019.02407.

Point 20: L367: Most ectomycorrhizal fungi also adopt saprophytic mode of nutrition. This is also true for pathogenic fungi (facultative).

Response 20: Many thanks for your professional comments. Yes, soil fungi of facultative trophic modes and functional guilds are prevalent. In this study, soil fungi were divided into trophic modes and functional guilds (including facultative functional guilds) by the FUNGuild database (http://funguild.org) [56]. It is worth noting that the EcM in L367 of the original manuscript refers specifically to obligate ectomycorrhizal fungi (Figure 2e). We have added more details for further studies on soil fungi with facultative trophic modes in the Discussion of the manuscript (clean).  Please see L531-L535 in the manuscript (clean) for further details.

Reference

  1. Nguyen, N.H.; Song, Z.; Bates, S.T.; Branco, S.; Tedersoo, L.; Menke, J.; Schilling, J.S.; Kennedy, P.G. FUNGuild: An open annotation tool for parsing fungal community datasets by ecological guild. Fungal Ecol. 2016, 20, 241-248, doi: 10.1016/j.funeco.2015.06.006.

Point 21: L390: Please recheck the data presented in figure 4.

Response 21: Thank you very much for your suggestion. We have rechecked the data presented in Figure 4 and found no data presentation errors.

Round 2

Reviewer 3 Report

The authors have taken into consideration all the suggested changes and revised the manuscript accordingly. Nevertheless, modify the title as "Altitudinal Variations Influences Soil Fungal Community Composition and Diversity in Alpine-Gorge Region on the Eastern Qinghai-Tibetan Plateau"

Author Response

Response to Reviewer 3 Comments

Point 1: The authors have taken into consideration all the suggested changes and revised the manuscript accordingly. Nevertheless, modify the title as "Altitudinal Variations Influences Soil Fungal Community Composition and Diversity in Alpine-Gorge Region on the Eastern Qinghai-Tibetan Plateau".

Response 1: Many thanks for your professional suggestion. We have modified the title as "Altitudinal Variation Influences Soil Fungal Community Composition and Diversity in Alpine-Gorge Region on the Eastern Qinghai-Tibetan Plateau". Please see L2-L4 in the manuscript (clean) for further details. Additionally, we have replaced “elevation” and “elevational” with “altitude” and “altitudinal” throughout the manuscript (clean) based on your suggestion. Please see that in the manuscript (clean) for further details.
